# Liquid Biopsy and Epigenetic Signatures in AML, ALL, and CNS Tumors: Diagnostic and Monitoring Perspectives

**DOI:** 10.3390/ijms26157547

**Published:** 2025-08-05

**Authors:** Anne Aries, Bernard Drénou, Rachid Lahlil

**Affiliations:** 1Institut de Recherche en Hématologie et Transplantation (IRHT), Hôpital du Hasenrain, 87 Avenue d’Altkirch, 68100 Mulhouse, France; ariesa@ghrmsa.fr (A.A.); drenoub@ghrmsa.fr (B.D.); 2Laboratoire d’Hématologie, Groupe Hospitalier de la Région de Mulhouse Sud-Alsace, Hôpital E. Muller, 20 Avenue de Dr. Laennec, 68100 Mulhouse, France

**Keywords:** epigenetic, liquid biopsy, methylome, circulating RNA, leukemia, lymphoma, PCNSL, early diagnosis

## Abstract

To deliver the most effective cancer treatment, clinicians require rapid and accurate diagnoses that delineate tumor type, stage, and prognosis. Consequently, minimizing the need for repetitive and invasive procedures like biopsies and myelograms, along with their associated risks, is a critical challenge. Non-invasive monitoring offers a promising avenue for tumor detection, screening, and prognostication. While the identification of oncogenes and biomarkers from circulating tumor cells or tissue biopsies is currently standard practice for cancer diagnosis and classification, accumulating evidence underscores the significant role of epigenetics in regulating stem cell fate, including proliferation, self-renewal, and malignant transformation. This highlights the importance of analyzing the methylome, exosomes, and circulating RNA for detecting cellular transformation. The development of diagnostic assays that integrate liquid biopsies with epigenetic analysis holds immense potential for revolutionizing tumor management by enabling rapid, non-invasive diagnosis, real-time monitoring, and personalized treatment decisions. This review covers current studies exploring the use of epigenetic regulation, specifically the methylome and circulating RNA, as diagnostic tools derived from liquid biopsies. This approach shows promise in facilitating the differentiation between primary central nervous system lymphoma and other central nervous system tumors and may enable the detection and monitoring of acute myeloid/lymphoid leukemia. We also discuss the current limitations hindering the rapid clinical translation of these technologies.

## 1. Introduction

Epigenetic modifications during cancer initiation and progression provide significant biological insights. Cancer cells primarily achieve these epigenetic changes through DNA methylation, histone modifications, remodeling and repositioning of nucleosomes, and circulating RNA (e.g., microRNA). The potential use of this knowledge in diagnosis and the design of more effective treatment strategies is currently a major focus in medical and scientific research. DNA methylation plays a fundamental role in cancer development and has emerged as a promising biomarker for various malignancies, including leukemia and brain tumors [1,2]. As a crucial epigenetic modification that regulates gene activity, it involves the addition of a methyl group to the cytosine base of DNA, specifically at sites known as CpG islands [3,4]. It is well documented that aberrant DNA methylation can lead to tumorigenesis and leukemia [1,5]. Abnormal DNA methylation occurs earlier in the cancerization process and is associated with the development of various malignancies. Early detection of dysregulated methylation may provide valuable insights into aberrant methylation patterns associated with malignancies, aiding in identifying cancer at an early stage, thus helping in crucial epigenetic-based therapeutic strategies [6,7]. In addition to methylation, circulating RNA enclosing a large group of RNA plays a significant role in complex cellular processes such as apoptosis, growth, differentiation, proliferation, and metabolism development. These RNAs could serve as excellent biomarkers for cancer and hematologic neoplasias detection [8]. Similarly to methylation, they are implicated in neoplastic initiation and progression, regulating many biological processes. This makes them highly advantageous as diagnostic biomarkers. Increasing studies confirm their contribution to neoplastic progression and their potential applications as novel biomarkers and therapeutic targets in cerebral cancer and leukemia treatment. Particularly, microRNA (miRNA) expression profiles have been used to classify tumors into different subtypes. Furthermore, circulating RNAs are present in biological fluids, inside extracellular vesicles (EVs) [9,10], making them stable non-invasive diagnostic and prognostic biomarkers or target of treatment for several diseases, including leukemia, lymphoma, and central nervous system (CNS) tumors. However, while the use of circulating RNA and the methylome to diagnose cancers is promising, further research and thorough validation of current studies are necessary. Researchers and clinicians have established that improving non-invasive and rapid diagnostic methods requires developing bioinformatics classification models that accurately compare one tumor type with others at the levels of differentially methylated regions (DMRs) of the methylome or specific circulating mRNA expression. While waiting for this powerful tool to become standard in cancer research and clinical practice, several challenges remain. These include ensuring reliability and reproducibility, as well as developing standardized analysis protocols across laboratories. Another challenge of liquid biopsy (LB) approaches in oncology is sensitivity due to the low abundance of materials in biological fluids such as serum and blood. The performance of these procedures models in testing and comparing sample sets is usually assessed in clinical biology by means of mean areas under receiver operating characteristic curves (AUROC). The discriminative power of key markers and their prognostic potentiality is typically evaluated using AUC and other statistical analyses. Models discerning tumor types generally achieve accuracy between 0.7 and 1.0. Unfortunately, for many tests, the results of positive and negative subjects often have an overlapping area, leading to classification errors. Some sick subjects may be classified as healthy and vice versa. The use of multiple markers and the development of machine learning-based medical methods capable of validating performance in diagnosing various diseases will be welcomed.

In summary, epigenetic analysis remains a powerful tool for diagnosing solid tumors and hematologic neoplasias, particularly leukemia, as well as neurological disorders and other syndromes. As technology advances, it is becoming a key component of precision medicine. This process requires only a few steps for disease detection and early diagnosis from LB by identifying specific circulating mRNA or cell-free DNA (cfDNA) methylation patterns from small samples of blood or other biological fluids (Figure 1). However, due to the individual variability, large-scale validation studies are needed. AI should be actively integrated into the workflow because of its remarkable performance in tasks such as data analysis, and addressing standardization issues, as methylation thresholds vary between laboratories.

## 2. Non-Invasive Diagnosis of Tumors and the Associated Challenges

Brain cancers are often challenging to diagnose due to their complex location and diverse symptoms. Primary central nervous system lymphoma (PCNSL) is a rare and aggressive type of lymphoma that affects the brain and can spread to the spinal cord, meninges, and eyes. In more than 90% of cases, the cancer cells are B lymphocytes. This type of lymphoma occurs more frequently in individuals with immune deficiencies. Therapeutic options have also evolved, with excellent results in younger patients while in older patients, they remain unsatisfactory requiring new options, of which several promising clinical trials are underway [11]. Distinguishing PCNSL from other brain cancers is challenging. However, brain tumors exhibit distinct DNA methylation profiles, which can help classify different subtypes and predict patient outcomes. Specific methylation patterns have been associated with brain cancers such as gliomas and medulloblastomas [12,13]. Tumor suppressor gene can be silenced due to promoter hypermethylation, promoting cancer growth. In contrast, hypomethylation of oncogenes may enhance their expression, contributing to tumor development. DNA methylation markers hold great promise for identifying and classifying various types of hematologic neoplasias and cancer, including brain cancer [14]. Similarly, circulating RNAs are being explored as potential biomarkers for the diagnosis and prognosis of cerebral tumors or lymphomas. Detecting methylated DNA and/or circulating RNA directly from biological fluids can aid in the early diagnosis, discriminating, and monitoring of brain tumors. On the other hand, leukemia and lymphomas are initiated primarily at the level of stem or progenitor cells that become cancerous following deregulation of genetic and epigenetic factors (e.g., methylation, circulating RNA) commonly expressed in normal stem cells. These hematologic neoplasias are frequently the result of genetic modifications and are characterized by the proliferation and maturation of neoplastic cells, leading to an increase in the number of transformed cells in the blood. It is believed that these hematologic diseases originate from the neoplastic transformation of stem cells or committed progenitor cells [15] through two types of events. First, normal stem cells can acquire genetic and epigenetic alterations that modify their growth control, increase their resistance to apoptosis, and interfere with cell differentiation. Second, non-stem cells can be modified by one or more oncogenes to reacquire the self-renewal properties of stem cells. In this context, determining the original cell, as well as the genetic and epigenetic events initiating blood disorders, can facilitate the detection, classification and treatment of these blood malignancies. Unfortunately, these events occur in organs such as bone marrow and lymph nodes, which cannot be easily handled and be biopsied extensively, requiring the search for less invasive and quicker detection methods. Analyzing circulating RNA or the methylome as valuable biomarkers through non-invasive procedures that can be repeated several times without side effects provides insights into specific epigenetic alterations associated with leukemia and lymphoma.

This approach aids in early detection, prognosis, and personalized treatment strategies for different cancers and hematologic neoplasias. Additionally, such analysis can be used as a method to define biomarkers for screening metastatic tumors, detecting cancer stages, assessing malignant progression, response to treatment, and detecting minimal residual disease.

### 2.1. DNA Methylation Screening from Liquid Biopsy as a Promising Biomarker for Early Tumor Diagnosis and Prognosis

#### 2.1.1. Acute Myeloid Leukemia

Acute myeloid leukemia (AML) is a hematologic malignancy that primarily affects older individuals and is characterized by the proliferation of immature cells, leading to cytopenias [16,17]. The cells of AML origin can also include committed progenitors lacking self-renewal capacity [18]. The standard and most effective treatment for AML typically involves intensive chemotherapy followed by allogeneic bone marrow transplantation. However, these treatments are often too aggressive for elderly patients, who may not tolerate the associated side effects. A significant clinical challenge is distinguishing between effective treatment and persistent disease activity in the weeks following therapy. Current management relies on regular bone marrow assessments via myelogram analysis to adjust treatment intensity, either by dose reduction or escalation. Ideally, a simple and reliable blood-based marker would allow for non-invasive and frequent monitoring of bone marrow status, quantifying indirectly the ratio of pathological to normal cells. To better understand disease progression, numerous studies have explored differential RNA expression and specific genetic mutations, aiming to identify genes associated with AML [19,20,21] and relapse [22]. However, substantial genetic heterogeneity among patients, even within specific AML subgroups, complicates the identification of universally applicable gene sets. In contrast, alterations in methylation patterns have been observed during AML initiation and progression. For instance, mutations in the methyltransferase *DNMT3A* lead to distinct methylation profiles associated with transcriptional variability of developmental and membrane-associated factors [23]. The current focus is on identifying potential epigenetic markers, differentially methylated in immature bone marrow cells of AML patients, but absent in normal cells. Such markers could be extended to the development of diagnostic assays for other malignancies.

Additionally, abnormal expression of *ALDH2* and *SPATS2L*, genes critical for AML patient survival, is correlated with DNA methylation at specific sites (*cg12142865* and *cg11912272*) [24]. This correlation underscores the potential of DNA methylation biomarkers for prognostic applications. In another study, three AML subtypes were identified based on methylation profiles and associated marker genes, offering potential targets for clinical therapy and precision medicine [25]. By investigating DNA methylation as a marker in cytogenetically normal acute myeloid leukemia (CN-AML), Cardoso et al. identified nine specific DNA sites (*cg23947872*, *cg12345678*, *cg87654321*, *cg23456789*, *cg98765432*, *cg34567890*, *cg09876543*, *cg45678901*, *cg10987654*) (Table 1) that can predict patient prognosis, independent of age [26]. These findings may facilitate the development of novel diagnostic tests and further studies to better categorize risk in this patient group. A separate study investigating DNA methylation alterations in AML patients identified differentially methylated regions and genes predictive of decitabine (DAC) response. Notably, GNAS, a PKA activator, emerged as a key DAC-responsive marker. Patients exhibiting *GNAS* hypomethylation demonstrated improved clinical outcomes following DAC treatment, suggesting *GNAS* as a potential prognostic biomarker for this therapy [27]. The methylation status of specific genes, such as GNAS, can therefore be utilized to predict patient outcomes and treatment responses, facilitating patient stratification and personalized treatment planning. Furthermore, Hua et al., employing digital PCR, demonstrated the potential of the increased methylation of *GRHL2* in AML patients as a biomarker for prognosis prediction and treatment response monitoring [28]. Additionally, *HOXA9* hypomethylation has been associated with diverse genetic abnormalities across AML subtypes. Notably, AML patients with *HOXA9* hypomethylation may derive greater benefit from transplantation, indicating that *HOXA9* methylation status could inform treatment decisions between transplantation and chemotherapy [29]. Moreover, elderly or intensive chemotherapy-unfit AML patients treated with 5-azacytidine (AZA) and the BCL2 inhibitor venetoclax (VEN) exhibited higher levels of *miR-182* promoter hypermethylation compared to normal controls [30]. These patients achieved faster complete remission and experienced prolonged overall and leukemia-free survival relative to those with hypomethylation. Consequently, *miR-182* promoter hypermethylation may serve as a valuable biomarker for predicting favorable outcomes in AML patients receiving AZA and VEN [30]. Furthermore, Schmutz et al., utilizing DNA methylation patterns, identified 1755 specific DMRs, including genes like *WNT10A* and *GATA-3*, to predict treatment response in AML patients undergoing AZA-containing induction therapy [31]. Analysis of these profiles enables improved patient risk stratification, leading to more precise and effective treatment strategies. Based on DNA methylation, termed CpG island methylator phenotypes (CIMP), Jian and colleagues identified three distinct AML subtypes, characterized by high, medium, and low DNA methylation levels, and can predict overall survival [32]. They developed a prognostic model based on 91 differential CpG sites and 32 key genes, enabling the prediction of survival rates from 0.5 to 5 years. This study highlights the potential of DNA methylation-based subtypes in improving AML prognosis and treatment [32]. Subsequently, the same group conducted a comprehensive analysis of gene mutations, methylation levels, mRNA expression, and AML-related genes from public databases in 103 AML samples, employing Weighted Gene Co-expression Network Analysis (WGCNA) [33]. This analysis revealed twelve key biomarkers associated with AML prognosis. A recent study by Guo et al. demonstrates that MLN4924 can reactivate tumor suppressor genes in AML by altering DNA methylation and gene expression [34]. Through a comprehensive analysis of DNA methylation and gene expression profiles in AML cells treated with the neddylation inhibitor MLN4924, they found that low *TRIM58* expression and high promoter methylation are associated with poor prognosis. This suggests that MLN4924 can modify DNA methylation and gene expression in AML, leading to the reactivation of tumor suppressor genes, including *TRIM58*. Additionally, by integrating multi-omics data from newly diagnosed AML patients and employing an unsupervised classification system, distinct subtypes were defined, enhancing the accuracy of survival prediction and guiding personalized treatment strategies [35]. Hypomethylation of *GCNT2* isoform A could predict poor survival and serve as a promising indicator to identify high-risk AML patients who might benefit from immunotherapy [36]. Furthermore, Dong and colleagues have recently used machine learning to uncover hypermethylation patterns in pediatric AML recurrence. Key hypermethylated genes identified include *SLC45A4*, *S100PBP*, *TSPAN9*, *PTPRG*, *ERBB4*, and *PRKCZ*, which are involved in crucial biological processes. These findings enhance our understanding of AML recurrence and offer potential pathways for improved prognostic models and treatments strategies [37]. Building on these insights, current research has extended the use of DNA methylation profiling to other hematologic malignancies such as lymphomas and acute lymphoblastic leukemia (ALL), where specific or aberrant methylation also plays a critical role in diagnosis, monitoring prognosis, and therapeutic decisions.

#### 2.1.2. Lymphoma and Acute Lymphoblastic Leukemia

The lack of reliable blood biomarkers for the early detection of lymphoma in patients is usually responsible for the diagnosis at advanced stages of the disease (stages III/IV) when it becomes perceptible with enlarged lymph nodes [38]. Acute lymphoblastic leukemia (ALL) is the most common pediatric hematologic neoplasias and consists of multiple subtypes with distinct abnormal gene expression profiles defined by groups of somatic mutations, chromosomal rearrangements, deregulating oncogenes or encoding chimeric fusion transcripts, and aneuploidy [39]. However, ALL cells are characterized by an unusually highly methylated genome and exhibit CpG island hypermethylation but minimal global loss of methylation [40], leading to the use of aberrant DNA methylation to characterize established subtypes and stratify risk groups of patients with ALL [41]. Thus, clinicians may generally consider multimodal diagnostic approaches when faced with the need to differentiate, for example, T-lymphoblastic lymphomas (T-LBL) that require intensive poly-chemotherapy and thymoma, because the prognosis and treatment are very different between them. Indeed, in such a case, it may eventually be necessary to resort to multimodal diagnostic approaches to establish the correct diagnosis in order to avoid wrongly exposing a patient to intensive chemotherapy in the event of a diagnostic error. Hence, Latiri et al. defined a classifier model based on six differentially methylated gene promoters able to accurately predict diagnoses of T-LBL and thymomas [42]. In another study, an assay was developed and tested in plasma-extracted and bisulfite-converted DNA targeting 16 cancer-specific methylated DNA markers from new untreated naive lymphoma patients and healthy controls. It allowed the detection of 78% (95% CI, 74–82%) of lymphoma cases with a specificity of 90% [43]. The identification of CpG islands in childhood ALL with the t(12;21) translocation, involving the *ETV6* and *RUNX1* genes, has revealed specific methylation of four genes, *DKK3*, *sFRP2*, *PTEN*, and *P73*. This pattern of genes simultaneously methylated, known as the CpG island methylator phenotype (CIMP), suggests that methylation could serve as a new biomarker for predicting risk in *ETV6/RUNX1*-postive ALL [44]. This finding could potentially impact treatment strategies and improve outcomes. The CIMP can enhance risk assessment in children with T-cell ALL by providing more accurate predictions of treatment outcomes. Thus, combining CIMP classification with measurable residual disease evaluation allows for better risk stratification in pediatric T-ALL, potentially improving patient outcomes [45]. A recent study on immature leukemia with ambiguous lineage, particularly those exhibiting a CIMP phenotype, found that these leukemias are more defined by their epigenetic profiles than by genetic mutations [46]. This epigenetic dysregulation accounts for the mixed phenotypes observed in AML, T-cell ALL, and mixed phenotype leukemia. Notably, this finding is especially relevant for ALL, where epigenetic changes can significantly impact disease progression and treatment response. The *VTRNA2-1* gene is a promising early detection biomarker, as its hypermethylation at birth has been associated with the subsequent development of pre-B-ALL. This hypermethylation is significantly associated with worse pre-B ALL patient survival and with its expression reduction. Indeed, DNA methylation alterations have been observed in newborns before the onset of pediatric precursor B-cell ALL (pre-B-ALL). *VTRNA2-1* methylation levels were higher at diagnosis, normalized at remission, and increased again at relapse, which correlates with poorer survival and could represent a target for epigenetic therapy [47]. Interestingly, a recent study developed a comprehensive cancer DNA methylation biomarker database (MethMarkerDB) for 13 types of cancer, including ALL [48]. The database provides significant insights that can enhance patient outcomes and advance ALL research. This resource identified 737,190 DMRs specific to ALL, with 719,088 hypermethylated and 18,102 hypomethylated [48]. These DMRs are crucial for understanding the epigenetic changes in ALL and can serve as potential biomarkers for diagnosis, prognosis and treatment. The growing evidence supporting the clinical utility of DNA methylation biomarkers in hematologic malignancies such as lymphoma and ALL highlights the broad potential of epigenetic profiling across diverse cancer types. As researchers continue to uncover methylation-based signatures that enhance diagnostic accuracy and enable personalized risk stratification, similar strategies are now being explored in solid tumors, including those of the central nervous system. In particular, the application of liquid biopsy and DNA methylation analysis offers promising avenues for non-invasive diagnosis and monitoring of brain tumors, where traditional biopsy approaches are often limited by invasiveness and risk.

#### 2.1.3. Central Nervous System Tumors

Cerebral primary tumors often include benign meningiomas, malignant gliomas, and lymphomas. When a brain tumor is suspected, accurately distinguishing between different types becomes a critical challenge. By using cerebrospinal fluid (CSF) and epigenetic markers, clinicians hope to avoid conventional brain biopsies and their associated risks. Liquid biopsy (LB) offers advantages over tissue biopsies and imaging approaches, providing significant cost and morbidity reductions compared to surgical excision of tissues and costly computed tomography (CT), positron emission, and magnetic resonance imaging (MRI) studies. Given that specific methylation patterns are associated with brain cancers, such as gliomas and medulloblastomas, DNA methylation markers hold great promise for detecting and classifying various types of tumors, including brain cancer [14]. Tumor suppressor genes can be silenced due to promoter hypermethylation, promoting cancer growth. Identifying differentially methylated DNA can aid in the early diagnosis and monitoring of brain tumors. Analyzing the methylome as a valuable biomarker provides insights into specific genetic alterations associated with brain cancers, aiding early detection, prognosis, and personalized treatment strategies. Once specific epigenetic alterations are identified and associated with a specific tumor; a simple digital PCR approach could be developed for classification from circulating CSF or plasma. Such methods could also be used as an alternative to targeting gene panels to identify recurrent hotspot mutations and copy number variations (CNVs) from CSF or plasma, addressing the difficulties encountered due to patient heterogeneities. Although the use of the methylome for diagnosing these cancers holds promise, further in-depth research and validation are necessary, along with advanced data treatment and bioinformatics, which currently hampers clinical application [49]. In clinical practice, differentiating between brain tumors such as glioblastoma (GBM), primary central nervous system lymphoma (PCNSL), and brain metastases (BMs) poses a significant diagnostic challenge. These tumors necessitate markedly different management approaches and treatment regimens. Due to the limitations of standard MRI in reliably distinguishing between these entities, invasive surgical biopsies are frequently required to establish definitive diagnoses and guide optimal therapeutic interventions. For example, Zuccato et al. used CSF-derived circulating tumor DNA methylation signature models to distinguish between malignant brain tumor types with a level of accuracy approaching that of standard-of-care (Table 1) [50]. Their method allowed the development of models that reliably distinguished between brain metastases (AUROC = 0.93, 95% confidence interval (CI): 0.71–1.0), glioblastomas (AUROC = 0.83, 95% CI: 0.63–1.0), and PCNSL (AUROC = 0.91, 95% CI: 0.66–1.0) in independent 20% validation sets [50]. Additionally, using the Methylation Outlier Detector program, other studies identified eight methylated markers that allowed the development of a new test to distinguish PCNSL from other CNS tumors [51] and the early detection and monitoring of cerebral B-cell lymphomas by following markers such as *HOXA9* and *GABRG3* [52]. These markers facilitate the development of biopsy-based tests to detect PCNSL-specific circulating tumor DNA, enhancing diagnostic accuracy and speed, ultimately improving patient outcomes. Recently, a model to predict brain metastases (BM) in lung cancer patients using DNA methylation patterns was developed. By analyzing the methylation profiles of 166 primary lung adenocarcinoma tumors, 5553 differentially methylated CpG sites were identified that significantly predict BM development. These sites were used to build and test a methylome-based model that accurately classifies patients into high, medium, or low risk for BM, potentially improving the management of lung cancer patients by enabling early prediction of BM development [53]. Moreover, by examining the molecular and cellular characteristics of IDH wild-type glioblastoma, an especially aggressive brain tumor, Lucas and colleagues developed an epigenetic evolution signature based on changes in DNA methylation at 347 critical CpG sites, showing a significant correlation with clinical survival outcomes [54]. These findings underscore the importance of incorporating genetic and epigenetic profiling into the clinical management of glioblastoma to enhance treatment outcomes. Specific DNA methylation markers unique to meningioma have been discovered in samples from 155 patients compared with other CNS tumors and non-tumor entities, highlighting that LB is a potential non-invasive and reliable tool for diagnosing and predicting outcomes in meningioma patients [55]. The ability to predict recurrence through these markers enables the stratification of patients based on their risk, facilitating more targeted follow-up and management strategies. Furthermore, Hinz et al. discovered a subgroup of high-grade IDH mutant astrocytromas with a primitive neuronal component and a unique DNA methylation profile, often mistaken for carcinoma metastasis [56]. Identifying this subgroup is crucial for accurate diagnosis, prognosis, treatment, and improving patient management and outcomes. Studies using multi-omics technologies have identified 178 prognostic genes, including 67 novel candidates, whose DNA methylation and gene expression are linked to glioma prognosis, offering new insights and highlighting potential targets for future research [57]. Interestingly, transcriptomics and epigenomic datasets with novel methylation DNA patterns were defined from formalin-fixed, paraffin-embedded primary brain tumors, including glioblastomas, astrocytomas, and oligodendrogliomas. This study included rare tumors, often neglected in brain cancer research, for comparison [58]. Furthermore, significant differences in DNA methylation patterns between long-term survivors and short-term survivors of World Health Organization grade II and III gliomas have been found, offering valuable insights for patient stratification and personalized treatment strategies [59]. These patterns may extend existing datasets with biomarkers from an independent, non-frozen cohort. By combining supervised and unsupervised learning approaches across nine transcriptomics datasets, several genes affected by DNA methylation in GBM (*ASPM*, *CCNB2*, *CDK1*, *AURKA*, *TOP2A*, *CHEK1*, *CDCA8*, *MCM10*, and *RAD51AP1*), have been identified, suggesting potential biomarkers for diagnosis and targets for therapy [60]. Overall, the integration of DNA methylation biomarkers into clinical practice holds great promise for improving the management of GBM through early detection, accurate prognosis, and personalized strategies.

**Table 1 ijms-26-07547-t001:** List of genes currently identified as potential biomarkers in different leukemia or CNS cancers.

Biomarker	Origin	Disease	Hypo/Hyper	Method	Ref.
*LDH2* and *SPATS2L*	bone marrow	AML	Hypo	TCGA/GEO	[24]
9 specific DNA sites (key genes: *UBE4A*, *MTMR1*, *ST6GALNAC1*, *CDK14*, *CA6*, *PDCD6IP*, *LCN6*, *FHL2*, *ITIH4*)	blood	AML	Hypo	WGBS	[26]
*GNAS*	bone marrow	AML	Hypo	WGBS	[27]
*GRHL2*	blood	AML	Hyper	ddPCR	[28]
*HOX9A*	bone marrow	AML	Hypo	TCGA, qMSP	[29]
*miR-182* promoter	bone marrow	AML	Hyper	B-Pyro-Seq qMSP	[30]
*WTN10A* and *GATA-3*	bone marrow	AML	Hyper	MeDIP-Seq qMSP	[31]
12 key biomarkers Hypo: *HOXB-AS3*, *HOXB3*, *MEG8* Hyper: *SLC9C2*, *CPNE8*, *S1PR5*, *MIR196B*	bone marrow	AML	Hypo/hyper	TCGA	[33]
*TRIM58*	cells	AML	Hyper	850K array, qMSP	[34]
*GCNT2*	Bone marrow	AML	hypo	TCGA/GEO	[36]
*SCL45A4*, *S100PBP*, *TSPAN9*, *PT*	Bone marrow	AML	hyper	GEO	[37]
6 promoters DMRs (*ZIC1*, *TSHZ2*, *CDC42BPB*, *RBM24*, *C10orf53*, *MACROD2*)	tissue	T-LBL/Thymomas	Hyper	EPICB array qMSP	[42]
*DKK3*, *sFRP2*, *PTEN*, and *P73*	bone marrow	childhood ALL	Hyper	qMSP, B-Seq	[44]
*CDKN2A*, *CDKN2B*, *PTEN*, *SHOX2*, *WT1*, *RASSF1A*, *TLX3*	bone marrow	ALL	Hyper	IE array	[45]
*CDKN2A*, *PTEN*, *SPI1*, *RUNX1*, *LEF1*, *CEBPA*	blood, bone marrow	ALL	Hyper	WGBS	[46]
*VTRNA2-1*	blood	Pre-B ALL	Hyper	IE array ddPCR	[47]
several biomarkers	CSF	CNS	hyper	IE array	[50]
*SCG3*, *NCOR2*, *KCNH7*, *DOCK1* cg05491001, cg25567674, *ZFPM2*, *GRIK1*	tissue, plasma	PCNLS	hyper	TAM-MSP assay	[51]
*HOXA9* and *GABRG3*	plasma	brain tumor	hyper	IE array qPCR	[52]
347 critical CpG sites (key genes: *MGMT*, *TERT*, *CDKN2A*, *PTEN*, *NF1...*)	tissue	GBM	110 hyper 153 hypo	IE array	[54]
*ASPM*, *CCNB2*, *CDK1*, *AURKA*, *TOP2A*, *CHEK1*, *CDCA8*, *MCM10*, *RAD51AP1*	tissue	GBM	hypo	TCGA	[60]

TCGA, the cancer genome atlas; GEO, gene expression omnibus; WGBS, whole-genome bisulfite sequencing; ddPCR, droplet digital PCR; qMSP, quantitative methylation-specific PCR; B-Pyro-Seq, bisulfite pyrosequencing; MeDIP-seq, methyl-CpG immunoprecipation sequencing; 850K array: Illumina Infinium MethylationEPIC BeadChip; EPICB array, Infinium MethylationEPIC BeadChip array; B-Seq, bisulfite sequencing; IE array: Illumina EPIC 450K array; TAM-MSP assay, Tailed Amplicon Multiplexed-Methylation-Specific PCA assay.

### 2.2. Exosomes and Circulating RNA as Biomarkers for Early Tumor Detection and Prognosis

Exosomes are small vesicles, ranging from 40 to 150 nm in diameter, enclosed by a lipid bilayer. They are secreted by most of cell types and carry a variety of cargo molecules, including lipids, proteins, specific cytokines, and circulating RNA (e.g., microRNAs, long non-coding RNAs, and circular RNA), making them valuable for diagnostic and therapeutic research [61,62]. Although issues of data standardization and analysis remain to be resolved, these molecules present differential expression profiles in cells as well as liquid biopsies during cancerization, making them interesting for non-invasive cancer diagnosis and prognosis. Thus, exploring these specific biomarkers to differentiate tumor subtypes may represent an epigenetic means and non-invasive methods for tumor screening and diagnosis.

Among circulating RNAs, miRNAs, which are small non-coding RNAs typically 18–25 nucleotides in length, can be found as free molecules or in exosomes. They play a crucial role in gene expression regulation and are implicated in tumor development due to their high presence in tumor tissues and their stability [63,64,65,66,67]. These miRNAs participate in numerous physiological processes, such as cell proliferation, apoptosis, and cancer development. They have been reported to regulate the expression of various oncogenes and tumor suppressor genes. Although most of miRNAs are produced and remain within cells, many are known as cell-free miRNAs and have been detected in various biological fluids, including blood and CSF [68]. Cell-free miRNAs, detectable in human blood and CSF, have been identified as potential markers for various diseases, including hematological malignacies and brain tumors, highlighting their significance in medical diagnostics and research [68]. The expression profiling of miRNAs is now used as diagnostics and prognostic biomarkers to evaluate tumor initiation, progression, and management in clinical oncology [69]. Determination of specific miRNA expression profiles may replace invasive biopsies and become a real advantage in the diagnosis of certain types of cancer and in predicting patient outcomes. Moreover, other circulating RNAs such as long non-coding RNAs (lncRNAs) and circular RNAs (circRNAs) have also been used as biomarkers in cancer diagnosis and prognosis. LncRNAs, which are longer than 200 nucleotides, regulate gene expression at multiple levels, thereby influencing cancer development and progression. This regulation accelerates the clinical implementation of lncRNAs as tumor biomarkers and therapeutic targets [70]. CircRNAs, on the other hand, act as molecular sponges for miRNAs, thereby affecting gene expression [71,72]. By interfering with miRNAs activity, circRNAs can modulate gene expression and impact various biological and pathological processes, including cancer development. Both lncRNAs and circRNAs have unique expression profiles that can serve as biomarkers for early cancer detection and prognosis [73]. The use of these non-coding RNAs, along with exosomes, offers a promising approach for non-invasive leukemia and cancer diagnostics and personalized treatment strategies.

#### 2.2.1. Acute Myeloid Leukemia

Epigenetic markers hold significant promise as diagnostic and prognostic tools for AML. MicroRNA (miRNA) expression profiles, such as *miR-155* and *miR-150*, can differentiate AML subtypes and serve as crucial biomarkers for disease progression and treatment outcomes [74]. The study of these miRNA expressions can detect AML relapses early through minimally invasive methods, easing timely treatment adjustments and potentially improving patient outcomes. Elevated levels of *miR-155* are associated with more aggressive leukemia forms and poorer prognosis [74,75]. Additionally, *miR-370* has shown superior diagnostic sensitivity compared to *miR-375*, making it a more effective non-invasive biomarker for pediatric AML [76]. Xia et al. identified over 20 types of small non-coding RNAs as reliable non-invasive biomarkers for AML to assist diagnosis and prognosis. Among these, *miR-181a*, which regulates gene involved in cell proliferation and differentiation, and *miR-155*, known for its role in immune response, show promise as AML biomarkers [77]. The *miR-182* directly targets *BCL2*, *PBX*, and *HOX9* mRNA, decreasing their protein expression through translational suppression in leukemic cells. This action facilitates leukemia stem cell self-renewal, accelerates AML progression, and influences the sensitivity to the BCL2 inhibitor Venetoclax [78,79]. Furthermore, Izadifard et al. identified a distinct panel of circulating plasma miRNAs, including *miR-638*, *miR-6511b-5p*, *miR-3613-5p*, *miR-455-3p*, *miR-5787*, and *miR-548a-3p* as promising non-invasive biomarkers for monitoring immune reconstitution following allogeneic hematopoietic stem cell transplantation in AML patients, highlighting their potential for post-transplant surveillance and personalized therapeutic strategies care [80].

lncRNAs also contribute to AML initiation, maintenance, and development by modulating differentiation, proliferation, cell cycle, and apoptosis [81]. Variations in lncRNA expression have been observed in various hematologic neoplasias, including AML. Recently, lncRNA deregulation has been recognized as a significant factor in AML progression, highlighting their potential as valuable prognostic markers [82]. Furthermore, by analyzing data from multiple studies, Salomon et al. demonstrated that lncRNAs exhibit high sensitivity and specificity in detecting AML [83]. Their ability to predict disease progression using non-invasive methods makes them valuable for personalized treatment strategies. Three lncRNAs (*XIST*, *TUG1*, and *GABPB1-AS1*) have been identified as key players in cytogenetically normal AML (CN-AML). Specifically, lncRNA *GABPB1-AS1* is highly expressed in CN-AML and negatively correlated with the overall survival, suggesting its potential as a prognostic biomarker and therapeutic target [84]. Other lncRNAs detected in various body fluids are suitable biomarkers for AML prognosis. Specific lncRNAs, such as *HOTAIR* (which promotes AML progression by regulating gene expression), *MALAT1A* (associated with poor prognosis), *MEG3* (a tumor suppressor often downregulated in AML), and *LINC00152* (implicated in AML cell proliferation and survivals), are crucial for AML prognosis [81,82,85], offering promising avenues for improving patient outcomes. Using the Cox method, Liu et al. established a regression model predicting AML prognosis with a reliable seven-lncRNA mapping tool (*LINC00461*, *RP11-309M23.1*, *AC016735.2*, *RP11-61I13.3*, *KIAA0087*, *RORB-AS1*, and *AC012354.6*) [86]. Recently, a study identified a 69-lncRNA signature, AML Lnc69, highly predictive of relapse risk in pediatric AML, refining current risk stratification and improving patient relapse prediction [87].

CircRNAs have also emerged as significant players in AML, due to their stability and specific expression in leukemic cells. Several circRNAs, such as *circRUNX1*, *cirWHSC1*, and *circFLT3*, are associated with poor clinical outcomes [88]. These circRNAs can be detected in blood samples, making them valuable for non-invasive follow-up and prognosis. Their presence holds significant prognostic value in pediatric AML, providing insights into disease progression and potential therapeutic targets. The growing body of evidence on the role non-coding RNAs in AML has significantly enhanced our understanding of its pathogenesis, disease monitoring, and treatment potential. Interestingly, similar mechanisms involving circulating RNAs are also being uncovered in other hematologic malignancies. In particular, acute lymphoblastic leukemia (ALL), the most common pediatric cancer, shows distinct patterns of non-coding RNA expression that offer promising avenues for early detection, prognosis, and therapeutic targeting.

#### 2.2.2. Acute Lymphoblastic Leukemia

Several studies have reported that miRNAs are emerging as important biomarkers in lymphoblastic leukemia, useful for diagnosis and prognosis. For instance, plasma levels of *miR-146a* are significantly elevated in ALL patients compared to healthy controls, highlighting its potential as a diagnostic biomarker [89]. Furthermore, *miR-146a* levels significantly decreased after treatment, indicating its usefulness in monitoring treatment response. These findings suggest that *miR-146a* could enhance the diagnosis and prognosis of ALL, providing a non-invasive method to track disease progression and treatment efficacy. Another study has shown that circulating *miR-128-3p* levels in blood, particularly in the exosome-enriched fraction, can serve as a promising non-invasive biomarker for monitoring minimal residual disease in childhood ALL [90]. Additionally, the expression of *miRNAs-181b-5p* is upregulated in ALL cell lines and enriched in blood vesicles and exosomes of patients cells, promoting cell proliferation and invasion, leading to poor relapse and outcomes [91]. Recent studies have shown that exosomal *miR-326* levels are significantly higher in patients cells with B-ALL compared to non-cancer controls, supporting its potential use as a non-invasive biomarker for diagnosing B-ALL [92]. Exosomes with elevated *miR-326* levels reduce the viability of drug-resistant B-ALL cells, suggesting that monitoring *miR-326* could help overcome drug resistance. These findings may contribute to advances in easier diagnosis and novel therapeutic strategies to overcome drug resistance in pediatric ALL, especially B-cell ALL, which remains a significant challenge. Moreover, research on the role of miRNAs as prognostic biomarkers in pediatric ALL assessed the expression levels of 84 miRNAs in bone marrow samples from newly diagnosed pediatric ALL patients at diagnosis and on day 33 of induction therapy [93]. In this study, specific miRNAs (*let-7c-5p*, *miR-106b-5p*, etc.) showed higher expression levels associated with a good prednisone response. Furthermore, differences in miRNA expression levels (*miR-125b-5p*, *miR-150-5p*, and *miR99a-5p*) were identified between standard/intermediate-risk and high-risk patients. These data suggest potential clinical implications for using these miRNAs in clinical practice for better risk stratification and personalized treatment [93]. However, larger studies are needed to confirm these results and further explore the therapeutic potential of targeting these miRNAs. Thus, many miRNAs expression levels in addition to their potential use in diagnosis are associated with drug resistance, predicting treatment outcomes and serving as a non-invasive diagnostic tool for primary drug resistance follow-up in pediatric ALL, thereby improving new therapeutic strategies for this type of cancer.

LncRNAs exhibit distinct expression profiles in ALL, allowing detection and differentiation between various subtypes of the disease. Specific lncRNAs are exclusively expressed in childhood B-cell acute lymphoblastic leukemia (B-ALL) and T-cell acute lymphoblastic leukemia (T-ALL), aiding in precise diagnosis. For example, Huang et al. determined a set of IncRNAs involved in different regulatory mechanisms differentially expressed in childhood ALL, associated with better leukemia-free survival and potentially serving as novel biomarkers to distinguish ALL subsets and outcome [94].

CircRNAs also hold promise for enhancing the diagnosis, and treatment of ALL. They influence key cellular processes, which are crucial for tumor invasion and progression. In ALL, specific circRNAs, like *circPVT1* and *circHIPK3*, enhance cell proliferation and survival, thereby accelerating tumor growth and serving as potential diagnostic and prognostic biomarkers [95,96]. Moreover, changes in circRNA expression, such as the upregulation of *circ-0000745*, can increase apoptosis by sponging *miR-193a*, aiding in the effective elimination of cancerous cells [97]. Comparisons of circRNA expression between human T-ALL and normal T-cell progenitor cells reveal different expression signatures in human T-ALL subtypes, suggesting their functional involvement in disease pathogenesis and highlighting their putative oncogenic or tumor suppressor roles [98]. In pediatric B-cell ALL, a circular RNA isoform of *WASHC2A* is notably upregulated in high-risk patients compared to healthy controls [99]. This circRNA is linked to poor prognosis, indicating its potential as a prognostic biomarker for high-risk B-ALL. The circRNA signature in childhood TCF3::PBX1 subtype B-cell ALL, has identified *cirANSK1B*, *cirBARD1*, and *cirMAN1A2* as overexpressed and potential biomarkers for prognosis and tumor detection. These findings offer valuable insights into the circRNA landscape of TCF3::PBX1 ALL, emphasizing their possible use for diagnosis and treatment [100]. While hematologic malignancies such as AML and ALL have been extensively studied in the context of circulating RNAs, these molecules are also gaining attention in solid tumors, especially those of CNS. Emerging research highlights the role of non-coding RNAs as valuable diagnostic and prognostic biomarkers in CNS tumors, offering novel insights into tumor biology and potential therapeutic targets in these challenging malignancies.

#### 2.2.3. Central Nervous System Tumors

Circulating RNAs play a significant role in many diseases and are being explored as potential biomarkers for the diagnosis and prognosis of cerebral tumors or lymphomas, as well as therapeutic targets. Elevated levels of *miR-10b*, *miR-130a*, and *miR-210* have been observed in glioma cells compared to normal brain tissues [101] (Table 2). Notably, *miR-10b* and *miR-210* are associated with glioma progression and poor diagnosis. Monitoring their altered expression could aid in early brain tumor diagnosis and guide treatment strategies [102]. A pioneering study by Baraniskin et al. revealed that the presence of *miR-21*, *mi-R19*, and *miR-92a* in CSF can aid in the diagnosis of PCNSL and help distinguish it from other CNS diseases [103]. These specific miRNAs provide valuable molecular insights into tumor characteristics, helping to differentiate between various types of brain tumors. Additionally, *miR-30c* has emerged as a novel diagnostic biomarker for both primary and secondary B-cell lymphoma of the CNS [104]. Stromek et al., using next-generation sequencing (NGS)-based miRNA profiling on CSF, identified a combination of four miRNAs (*miR-16-5p*, *miR-21-5p*, *miR-92a-3p*, and *miR-423-5p*) that can discriminate PCNSL from non-malignant samples [105]. These miRNAs have a significant potential for non-invasive diagnosis of PCNSL, possibly leading to earlier and more accurate detection. On the other hand, *miR-124* is considered a diagnostic and prognostic biomarker for glioma, as it is associated with molecules involved in crucial cellular processes, making it a precise indicator of cellular health and disease [106]. In comparison, other miRNAs such as *miR-21* and *miR-221* are also used as biomarkers, but may not offer the same level of specificity or consistency in gliomas diagnosis [107]. A recent study investigating the role of specific circulating miRNAs as biomarkers for predicting prognosis and treatment response in glioblastoma revealed the upregulation of *miR-29a*, *miR-106a*, and *miR-200* in GBM patients compared to healthy individuals [108]. The expression levels of these three miRNAs were significantly reduced post-treatment, indicating their potential use as markers for monitoring treatment efficacy. By integrating both supervised and unsupervised learning approaches across nine transcriptomics datasets, several miRNAs, including *miR-16-5p*, *miR-34a-5p*, *miR-205-5p*, *miR-124-3p*, and *miR-147a* have been identified as playing significant roles in GBM diagnosis [60]. Bustos et al. identified *miR-3180-3p* and *miR-5739* as potential biomarkers for the identification of primary and recurrent GBM. These cell-free RNAs revealed high specificity and sensitivity in differentiating glioblastoma patients from healthy individuals [109]. These findings suggest that these circulating miRNAs have potential use in diagnosis and disease monitoring and could provide valuable insights for personalized therapeutic strategies in glioblastoma management. In addition to miRNA, several lncRNAs, including *HOTAIR*, *MALAT1*, *TUG1*, and *NEAT1*, play crucial roles in regulating gene expression and significantly impact tumor progression [110]. These lncRNAs promote cell proliferation, migration, and therapy resistance, making them potential biomarkers for early detection, diagnosis, prognosis, and therapeutic targets in malignant brain tumors. A recent study found that *SLCO4A1-AS1*, an lncRNA driven by super-enhancers, is significantly upregulated in GBM compared to non-tumor tissues. This upregulation correlates with older patient age, poor prognosis, and higher-grade tumors, making *SCLO4A1-AS1* a potential biomarker for GBM [111]. Targeting *SCL04A1-AS1* could be a valuable therapeutic strategy for GBM, potentially enhancing the effectiveness of existing treatments, such as the ERK inhibitor VX-11e, and providing new therapeutic avenues [111]. Another study by the same group demonstrated that *ZNF503-AS2* expression is significantly upregulated in glioblastoma compared to normal brain tissues. *ZNF503-AS2* knockdown reduces proliferation, invasion, and migration as well as promote apoptosis [112]. Therefore, *ZNF503-AS2* may offer new opportunities for research and therapy development as a potential therapeutic target for glioma treatment. Moreover, the expression levels of both *LINC00565* and *LINC00641* were also significantly upregulated in GBM patients compared to healthy controls. These elevated levels correlated with larger tumor size, poor performance status, and worse progression-free survival and overall survival [113]. Collectively, these studies support the potential of lncRNAs as valuable biomarkers for glioma, highlighting their importance in diagnosis, prognosis, and as therapeutic targets. Among potential circRNAs serving as biomarkers for various cancers, *CircHIPK3* has been identified as a sponge for *miR-124*, thereby promoting cell proliferation and invasion. Its expression levels can be used as both diagnostic and prognostic biomarkers [114,115]. Additionally, *circSMARCA5*, an upstream regulator of *miR-126-3p* and *miR-515-5p* miRNA expression and their downstream targets, *IGFBP2* and *NRAS* mRNAs in GBM cells, is known to be dysregulated in GBM [67,116]. It is identified as a tumor suppressor and, like *circHIPK3*, can be used for both diagnosis and prognosis for glioblastoma multiforme [116]. Finally, *CircFBXW7*, noted for its tumor-suppressive functions in gliomas, can be used to detect tumor presence and predict patient outcomes [114,117]. These examples underscore the significant potential of circRNAs as versatile biomarkers in brain cancer diagnosis, paving the way for more personalized and effective therapeutic strategies.

**Table 2 ijms-26-07547-t002:** List of the main circulating RNAs identified as potential biomarkers in different Leukemia or CNS tumors.

Biomarker	Type	Origin	Disease	Hypo/Hyper	Method	Ref.
*miR-155*, *miR-150*	miRNA	Serum exosomes, EVs	AML	up	TLDA qRT-PCR	[74,75]
*miR-370*	miRNA	blood	AML	down	qRT-PCR	[76]
*miR-181a*, *miR-155*	miRNA	serum, bone marrow	AML	up	NGS qRT-PCR	[77]
*miR-182*	miRNA	cell lines	AML	down	qRT-PCR	[78,79]
*miR-548a*, *miR-6511b*, *miR-455*, *miR-5787*, *miR-638*, *miR-3613*	miRNA	plasma	AML	up down	qRT-PCR	[80]
*HOTAIR*, *MALAT1A*	lncRNA	bone marrow	AML	up	RNA-Seq qRT-PCR	[81,82,85]
*MEG3*	lncRNA	cells, bone marrow	AML	down	RNA-Seq qRT-PCR	[81,82,85]
*LINC00152*	lncRNA	cell lines, bone marrow	AML	up	RNA-Seq qRT-PCR	[81,82,85]
*XIST*, *TUG1*, *GABPB1-AS1*	lncRNA	cell lines	CN-AML	up	RNA-Seq TCGA	[84]
*LINC00461*, *RP11-309M23.1*, *AC016735.2*, *RP11-61I13.3*, *KIAA0087*, *RORB-AS1*, and *AC012354.6*	lncRNA	bone marrow	AML	up	RNA-Seq	[86]
*69-lncRNA*	lncRNA	bone marrow	AML	up	RNA-Seq	[87]
*circRUNX1*, *cirWHSC*, *circFLT3*	circRNAs	bone marrow	AML	up	RNA-Seq	[88]
*miR-146a*	miRNA	plasma	ALL	up	qRT-PCR	[89]
*miR-128-3p*	miRNA	blood	ALL	up	qRT-PCR	[90]
*miRNAs-181b-5p*	miRNA	blood	ALL	up	qRT-PCR	[91]
*miR-326*	miRNA	exosome	ALL	up	qRT-PCR	[92]
*miR-125b-5p*, *miR-150-5p*, *miR99a-5p*	miRNA	Bone marrow	ALL	down	qRTP-CR	[93]
*TCONS_00026679*, *uc002ubt.1*, *ENST00000411904*, *ENST00000547644*	lncRNA	Bone marrow	ALL	down	RNA-Seq	[94]
*circPVT1*, *circHIPK3*	circRNA	Cells lines	ALL	up	qRT-PCR	[95,96]
*circ-0000745*	circRNA	bone marrow, cell lines	ALL	up	qRT-PCR	[97]
*circWASHC2A*	circRNA	bone marrow	ALL	up	qRT-PCR	[99]
*circANSK1B*, *CircBARD1*, *cirMAN1A2*	circRNA	Bone marrow	ALL	up	RNA-Seq	[100]
*miR-10b*, *miR-130a*, *miR-210*	miRNA	serum	glioma	up	Databases (Scopus, Pubmed)	[101]
*miR-21*, *mi-R19*, and *miR-92a*	miRNA	CSF	PCNSL	up	qRT-PCR	[103]
*miR-30c*	miRNA	CSF	SCNLS	up	qRT-PCR	[104]
*miR-16-5p*, *miR-21-5p*, *miR-92a-3p*, *miR-423-5p*	miRNA	CSF	PCNSL	up	qRT-PCR	[105]
*miR-124*	miRNA	serum, plasma, tissue	glioma	down	-	[106]
*miR-21*, *miR-221*	miRNA	CSF, serum	glioma	up	-	[107]
*miR-29a*, *miR-106a*, *miR-200*	miRNA	blood	GBM	up	qRT-PCR	[108]
*miR-16-5p*, *miR-34a-5p*, *miR-205-5p*, *miR-124-3p*, and *miR-147a*	miRNA	CSF	GBM	down	TCGA-GEO	[60]
*miR-3180-3p*, *miR-5739*	miRNA	plasma	GBM	up	NGS qRT-PCR	[109]
*HOTAIR*, *MALAT1*, *TUG1*, *NEAT1*	lncRNA	CSF, serum	glioma, GBM	up	-	[110]
*SLCO4A1-AS1*	lncRNA	tissue	GBM	up	RNA-Seq	[111]
*ZNF503-AS2*	lncRNA	tissue	GBM	up	RNA-Seq qRT-PCR	[112]
*LINC00565*, *LINC00641*	lncRNA	blood	GBM	up	qRT-PCR	[113]
*CircHIPK3*	cirRNA	tissue, cells	Glioma	up	qRT-PCR	[114,115]
*CircHIPK3*, *circSMARC5*	cirRNA	serum	GBM	up	qRT-PCR	[67,116]
*CircFBXW7*	cirRNA	tissue	Glioma	down	RNA-Seq	[114,117]

TLDA, TaqMan Low-Density Arrays; qRT-PCR, quantitative reverse transcriptase PCR; NGS, Next-generation sequencing to profile the global expression of circulating small non-coding RNAs; GEO, gene expression omnibus; TCGA, the cancer genome atlas.

## 3. Prospect and Future Direction

The use of methylated DNA or circulating mRNA as markers represents a significant advancement in the diagnosis and management of brain tumors, offering a less invasive and faster alternative to traditional methods, facilitating both treatment and subsequent surveillance (Figure 2). Brain cancers are often challenging to diagnose due to their complex anatomical location and diverse clinical manifestations. These challenges are compounded by the intricacies of the brain and the reliance on traditionally invasive diagnostic methods. Although imaging techniques such as MRI and CT scans are routinely employed, they often lack the precision required to distinguish between benign and malignant tumors. Improving the classification of brain tumors and enabling the diagnosis and distinction of PCNSL, which is considered as hematologic cancer, from other central nervous tumors through a simple LB of CSF combined to epigenetic could lead to faster, less invasive diagnostic procedures simplifying decision-making for effective treatment. Similarly, the study of epigenetic modifications such as DNA methylation alterations or circulating RNA expression that occur during the initiation and progression of leukemia holds promise for improving diagnosis, monitoring disease progression, and potentially reducing the need for repeated bone marrow biopsies.

Currently, diagnostic methods for CNS cancers rely heavily on conventional MRI, which frequently necessitates additional invasive biopsies to confirm the diagnosis and guide treatment. In the context of PCNSL, liquid biopsy has previously been employed to detect specific mutations in tumor DNA isolated from plasma [118]. For instance, the L265P mutation in the *MYD88* gene has been used as a molecular marker to detect disease presence in cfDNA from CSF biopsies [119,120]. However, the absence of recurrent mutations in all tumors limits the universal applicability of this method, highlighting the methylome and circulating RNA approaches. Recent advances in molecular diagnostics offer hope for more accurate and less invasive detection through LB, particularly by identifying epigenetic changes associated with brain tumors, hematologic malignancies, or with other cancers [121,122,123]. Unlike traditional biopsies, which carry risks and discomfort, LB and epigenetic offer a safer, faster, and more accessible alternative to regular screening (Figure 2). Early detection through LB could significantly enhance patient outcomes and reshape conventional diagnostic paradigms.

Brain tumors display distinct DNA methylation signatures that can support sub-classification and prediction of clinical outcomes. Moreover, methylation profiling of cfDNA in CSF can help determine the cellular origin of the tumor and distinguishing, for example, between glial or B-cell lymphoid origins by determining the methylation status of tissue-specific genes. This approach could provide a more comprehensive molecular understanding of the tumor identified on imaging. Such insights are increasingly recognized as critical for guiding diagnosis and tailoring therapeutic strategies, making them a key focus in contemporary oncological and neurological research. Epigenetic markers such as DNA methylation patterns and circulating RNA signatures are dynamic and responsive to therapeutic interventions, making them promising candidates for real-time disease tracking. Serial epigenetic assessments could enable clinicians to non-invasively monitor treatment response, detect minimal residual disease, and predict relapse, thereby reducing reliance on invasive procedures such as biopsies and myelograms. Incorporating these tools into follow-up protocols could support a more adaptive and personalized approach to cancer management.

Nonetheless, while DNA methylation profiling holds great promise for classifying CNS tumors with high precision, some tumors remain unclassifiable. This can delay treatment initiation, particularly in aggressive cancers such as glioblastomas [124]. Most current studies use PCR-based methods to assess promoter methylation, often assuming a linear relationship between methylation and clinical outcomes. However, research into the O6-methylguanine-DNA methyltransferase (*MGMT*) promoter methylation in GBM suggests a nonlinear relationship with patient survival. Unusually, patients with intermediate methylation levels showed the best survival rates, followed by those with high levels, whereas low methylation was associated with increased mortality [125]. These findings underscore the need for quantitative rather than binary assessments of methylation to improve prognostic accuracy. Additionally, analyzing circulating mRNA and its altered expression profiles could uncover key molecules involved in disease biology, prognosis, and drug resistance in leukemia and CNS malignancies. On the other hand, several limitations are associated with these emerging diagnostic techniques. One major constraint is low tumor purity, as tumor ctDNA represents only a small fraction of plasma (only 1% are tumor ctDNA), which can impair the accuracy of classification methods, including methylation profiling, due to contamination by non-tumor cells [126]. Addressing tumor heterogeneity through single-cell resolution techniques is therefore gaining attention. Another fundamental question in epigenetics remains to determine if these modifications are a cause or a consequence of malignancy. Understanding this causal relationship is essential for establishing reliable biomarkers in clinical oncology. Furthermore, despite the growing enthusiasm surrounding liquid biopsy and epigenetic profiling, other significant challenges remain. Many of these methods still lack standardization, and their clinical implementation is hindered by variability in sensitivity, specificity, and reproducibility across studies. A key technical issue is inter-laboratory variability, which arises from differences in assay platforms, sample handling, and data normalization. These inconsistencies can significantly affect reproducibility and hinder cross-study comparisons [127]. Signal variation is another major concern, particularly for low-abundance circulating biomarkers such as cell-free DNA, RNA, or methylated fragments. Factors like sample degradation, background noise, and platform sensitivity can compromise diagnostic accuracy and limit clinical utility [128]. Moreover, the biological complexity and variation in tumors, particularly their heterogeneity and evolving epigenetic landscapes, raises concerns about the consistency and reliability of these biomarkers over time and across patient populations. Without rigorous validation in large, prospective clinical trials, their utility may remain limited to research settings. Regulatory and standardization hurdles remain substantial. The lack of universally accepted protocols, clinical-grade bioinformatics pipelines, and quality control frameworks delays the transition of these tools from bench to bedside through clinical endorsements. Regulatory approval processes require robust clinical validation and reproducibility, which many emerging assays have yet to demonstrate. Until these barriers are systematically addressed, the integration of liquid biopsy and epigenetic profiling into routine clinical workflows will remain aspirational rather than operational. Therefore, while these approaches hold transformative potential, their translation into routine clinical practice must be approached with caution and grounded in robust scientific evidence. Finally, regulatory hurdles remain a significant bottleneck. Biomarkers must undergo rigorous analytical and clinical validation to meet regulatory standards. The lack of harmonized guidelines, especially for epigenetic and non-coding RNA biomarkers, further delays their clinical adoption [129]. To address these challenges, future efforts should prioritize standardized protocols, multi-center validation, and integrative biomarker panels that combine genomic, epigenomic, and transcriptomic data. These strategies are essential for improving the robustness, reproducibility, and clinical relevance of emerging biomarkers.

## 4. Conclusions

To optimize epigenetic diagnostics, it is essential to enhance sensitivity, specificity, and cost-effectiveness while ensuring clinical feasibility. This includes identifying and validating highly specific biomarkers, exploring other epigenetic modifications such as cancer-specific histone marks, and profiling circulating RNA molecules. Moreover, the complexity of the data necessitates sophisticated bioinformatics tools and expert interpretation. The integration of artificial intelligence (AI) and machine learning is increasingly essential for outcome classification and pattern recognition. Training AI models to detect subtle epigenetic variations can aid in distinguishing benign from malignant profiles. Finally, widespread clinical adoption of these techniques requires the standardization, validation, and harmonization of epigenetic tests to ensure reproducibility across laboratories and healthcare systems.

## Figures and Tables

**Figure 1 ijms-26-07547-f001:**
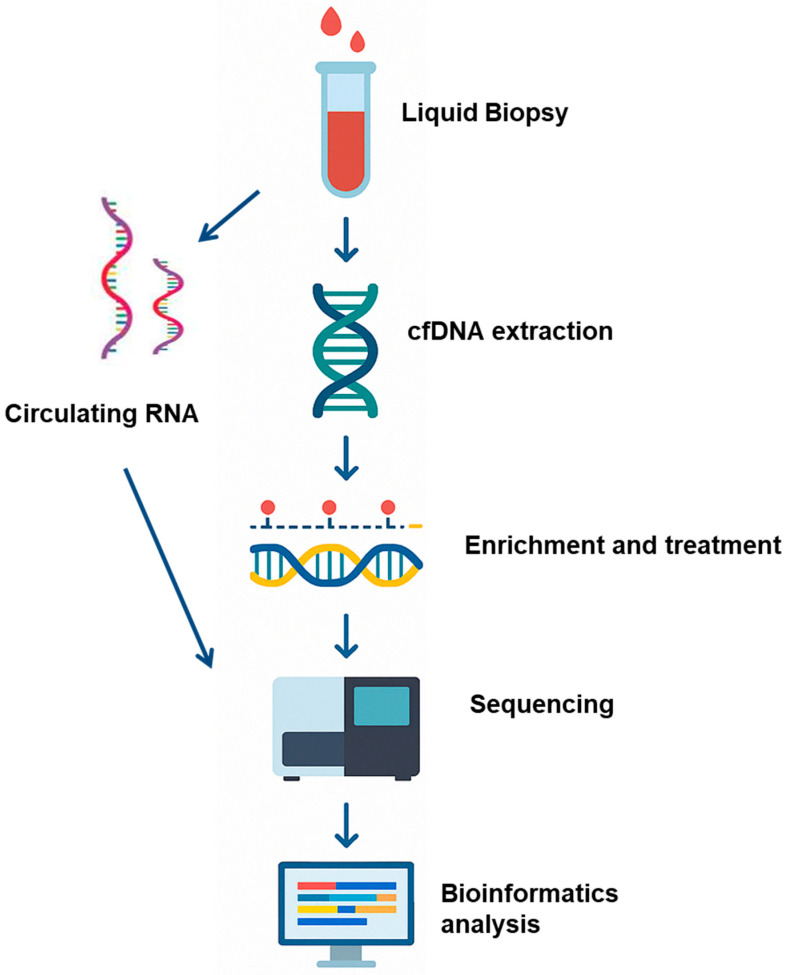
Liquid biopsy is combined with epigenomic profiling as a biomarker approach for cancer detection and monitoring. Schematic illustration of experimental and bioinformatics procedures: Blood samples or cerebrospinal fluid are collected from cancer patients or non-cancer control donors. Circulating RNA or cfDNA is then extracted from the participant’s sample, and the cfDNA is subjected to cytosine conversion and NGS analysis. Transcriptomic profiling of circulating mRNA and quantification of methylation-specific signals (e.g., methylated fragment ratios) are performed. A multimodal ensemble classifier integrates molecular features from both transcriptomic and epigenomic datasets to generate a probabilistic score indicating the presence of malignant transformation.

**Figure 2 ijms-26-07547-f002:**
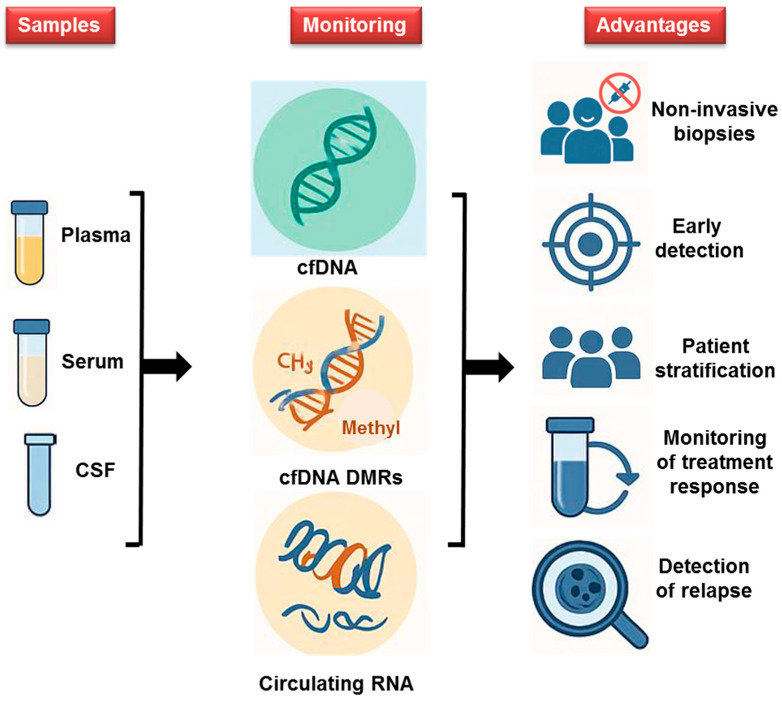
Innovative Diagnostics for Leukemia and Brain Tumors: The use of liquid biopsies (plasma, serum, and CSF), combined with methylated DNA and circulating mRNA as epigenetic markers for diagnosing leukemia (ALL and AML) and brain tumors, offers less invasive, faster alternatives to traditional diagnostics. This approach improves early diagnosis and detection of relapse and treatment decisions, ultimately enhancing patient outcomes.

## Data Availability

No new data were created or analyzed in this study. Data sharing is not applicable to this article.

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
