# Peer review of "Liquid Biopsy and Epigenetic Signatures in AML, ALL, and CNS Tumors: Diagnostic and Monitoring Perspectives"

_ijms, 2025, doi:10.3390/ijms26157547_

Round 1

Reviewer 1 Report

Comments and Suggestions for Authors

This review provides a comprehensive overview of the role of liquid biopsy and epigenetic regulation. The authors discuss the integration of the two approaches to facilitate rapid and efficient early detection and monitoring of leukemias and CNS tumors. The manuscript is overall well written and successfully underscores the need for clinical implementation of minimally invasive diagnostic and monitoring tools. Discussion of both DNA and RNA based methylation profiling represents an important component of this review.

However, the manuscript would benefit from clearer organization and refinement for a more cohesive narrative. The transitions between sections on leukemia, lymphoma, and CNS tumors could be strengthened to better differentiate shared versus disease-specific insights.

Additionally, while the review summarizes many individual biomarkers, their relative clinical readiness (preclinical, early validation, clinical trial, or approved test) is not always clear. Inclusion of a table outlining the clinical development stage (e.g. FDA approval, breakthrough device designation) and performance metrics (sensitivity, specificity) of the most promising markers would enhance the utility of the review.

Lastly, more critical discussion of the technical limitations (inter-laboratory variability, signal variation) disease specific challenges (tumor heterogeneity, varying response to targeted therapeutics) and regulatory hurdle would strengthen the relevance.

Overall, this is a valuable contribution to the field of liquid biopsy that highlights the growing clinical potential of epigenetic and transcriptomic profiling in liquid biopsy. With some structural and clarity revisions, this review could serve as a useful reference for critically analyzing non-invasive diagnostics in leukemia and CNS malignancies.

Author Response

Reviewer 1 :

Comments and Suggestions for Authors

This review provides a comprehensive overview of the role of liquid biopsy and epigenetic regulation. The authors discuss the integration of the two approaches to facilitate rapid and efficient early detection and monitoring of leukemias and CNS tumors. The manuscript is overall well written and successfully underscores the need for clinical implementation of minimally invasive diagnostic and monitoring tools. Discussion of both DNA and RNA based methylation profiling represents an important component of this review.

However, the manuscript would benefit from clearer organization and refinement for a more cohesive narrative. The transitions between sections on leukemia, lymphoma, and CNS tumors could be strengthened to better differentiate shared versus disease-specific insights.

We appreciate your feedback regarding the organization and narrative flow of the manuscript. In response, we have revised the structure to enhance coherence and improve transitions between the sections on leukemia, lymphoma, and CNS tumors. We hope these changes result in a more cohesive and integrated narrative, in line with your recommendation. All modifications and additions are highlighted in red in the revised version of the manuscript.

Additionally, while the review summarizes many individual biomarkers, their relative clinical readiness (preclinical, early validation, clinical trial, or approved test) is not always clear. Inclusion of a table outlining the clinical development stage (e.g. FDA approval, breakthrough device designation) and performance metrics (sensitivity, specificity) of the most promising markers would enhance the utility of the review.

Thank you for this valuable suggestion. We agree that a table summarizing the clinical development stages and performance metrics of the most promising biomarkers would significantly enhance the utility of the review. However, the currently available data for many of the biomarkers discussed remains limited or incomplete, particularly regarding standardized performance indicators such as sensitivity, specificity, and regulatory status. As such, compiling a comprehensive and accurate table for most of these markers is not feasible currently without risking misrepresentation.

Nevertheless, we previously attempted to address this by including specific examples within certain sections in the initial version of the review. To further highlight this issue, we have now added a discussion at the end of the paragraph titled “3. Perspectives and Future Directions” outlining the current progress and state of clinical validation for the biomarkers mentioned. We fully recognize the importance of this point and acknowledge that it will require regular updates as the field evolves and new validation studies become available. We hope to return to this in a dedicated future study.

Lastly, more critical discussion of the technical limitations (inter-laboratory variability, signal variation) disease specific challenges (tumor heterogeneity, varying response to targeted therapeutics) and regulatory hurdle would strengthen the relevance.

Thank you for your valuable advice. We agree that a more critical discussion of the technical and disease-specific limitations would enhance the manuscript’s relevance. Accordingly, we have expanded the discussion section to address these points in Section “3. Perspectives and Future Directions”. As a result, three references have been added (128, 129 and 130).

We believe these adjustments and additions strengthen the manuscript by providing a more comprehensive perspective on both the translational potential and the limitations of current research in the field.

Overall, this is a valuable contribution to the field of liquid biopsy that highlights the growing clinical potential of epigenetic and transcriptomic profiling in liquid biopsy. With some structural and clarity revisions, this review could serve as a useful reference for critically analyzing non-invasive diagnostics in leukemia and CNS malignancies.

Reviewer 2 Report

Comments and Suggestions for Authors

Review Report

Title: Liquid Biopsy and Epigenetic Regulation for Early Detection and Monitoring of Leukemia and CNS tumors

Authors: Anne Aries, Bernard Drénou and Rachid Lahlil.

Comments:

In their paper “Liquid Biopsy and Epigenetic Regulation for Early Detection and Monitoring of Leukemia and CNS tumors”. Aries and colleagues clarify the importance of diagnostic assays that integrate liquid biopsiesc with epigenetic analysis for revolutionizing tumor management by enabling rapid, non-invasive diagnosis, real-time monitoring, and personalized treatment decisions.

The article is a well-structured review summarizing current studies which investigated the use of epigenetic regulation, specifically the methylome and circulating RNA, as diagnostic tools derived from liquid biopsies. In addition, they underscore how this is approach could facilitate the differentiation between primary central nervous system lymphoma and other central nervous system tumors and enables the detection and monitoring of acute myeloid/lymphoid leukemia. Furthermore, they discuss the current limitations hindering the rapid clinical translation of these technologies.

Taken together, the study adds substantial new knowledge regarding epigenetic analysis as a powerful tool for diagnosing cancers, particularly leukemia and neurological disorders. They found that artificial intelligence (AI) and machine learning should be actively integrated into the workflow exploring the epigenetic variations. This could have considerable clinical impact by helping in distinguishing benign from malignant profiles and enhancing sensitivity, specificity, and cost-effectiveness of epigenetic tests

The authors should, however add some minor additional modifications to the manuscript:

  • The title of the article needs to be modified as the review is not covering information about all types of leukemia or CNS tumors. Therefore, the title is considered broad and needs to be more specified about the topics discussed in the article.

  1. The sentence starts in line 64 and ends in line 67 is very long and this makes it harder to be understood. It is better to make this sentence shorter and clearer.

Author Response

Reviewer 2 :

Title: Liquid Biopsy and Epigenetic Regulation for Early Detection and Monitoring of Leukemia and CNS tumors

 Authors: Anne Aries, Bernard Drénou and Rachid Lahlil.

 Comments:

 In their paper “Liquid Biopsy and Epigenetic Regulation for Early Detection and Monitoring of Leukemia and CNS tumors”. Aries and colleagues clarify the importance of diagnostic assays that integrate liquid biopsiesc with epigenetic analysis for revolutionizing tumor management by enabling rapid, non-invasive diagnosis, real-time monitoring, and personalized treatment decisions.

 The article is a well-structured review summarizing current studies which investigated the use of epigenetic regulation, specifically the methylome and circulating RNA, as diagnostic tools derived from liquid biopsies. In addition, they underscore how this is approach could facilitate the differentiation between primary central nervous system lymphoma and other central nervous system tumors and enables the detection and monitoring of acute myeloid/lymphoid leukemia. Furthermore, they discuss the current limitations hindering the rapid clinical translation of these technologies.

 Taken together, the study adds substantial new knowledge regarding epigenetic analysis as a powerful tool for diagnosing cancers, particularly leukemia and neurological disorders. They found that artificial intelligence (AI) and machine learning should be actively integrated into the workflow exploring the epigenetic variations. This could have considerable clinical impact by helping in distinguishing benign from malignant profiles and enhancing sensitivity, specificity, and cost-effectiveness of epigenetic tests

 The authors should, however add some minor additional modifications to the manuscript: 

  • The title of the article needs to be modified as the review is not covering information about all types of leukemia or CNS tumors. Therefore, the title is considered broad and needs to be more specified about the topics discussed in the article.

Thank you for your observation. We have revised the title of the article to more accurately reflect the specific types of leukemia and CNS tumors discussed in the review. The updated title is: Liquid Biopsy and Epigenetic Regulation for Early Detection and Monitoring of AML, ALL and CNS tumors”

  1. The sentence starts in line 64 and ends in line 67 is very long and this makes it harder to be understood. It is better to make this sentence shorter and clearer. 

We agree that the sentence from lines 64 to 67 is too long and may affect readability. We have revised it to make it shorter and clearer, as follows:

"While waiting for this powerful tool to become standard in cancer research and clinical practice, several challenges remain. These include ensuring reliability and reproducibility, as well as developing standardized analysis protocols across laboratories."